# Changing Treatment Philosophy of Slipped Capital Femoral Epiphysis (SCFE) after Introduction of the Modified Dunn Procedure (MDP): Our Experience with MDP and Its Complications

**DOI:** 10.3390/children10071163

**Published:** 2023-07-03

**Authors:** Enrico Micciulli, Laura Ruzzini, Giulio Gorgolini, Pier Francesco Costici, Fernando De Maio, Ernesto Ippolito

**Affiliations:** 1Department of Orthopaedic Surgery, Bambino Gesù Hospital, 00165 Rome, Italy; enrico.micciulli@opbg.net (E.M.); laura.ruzzini@opbg.net (L.R.); costici@libero.it (P.F.C.); 2Department of Orthopaedic Surgery, University of Rome “Tor Vergata”, 00133 Rome, Italy; g.gorgolini@gmail.com (G.G.); demaio@med.uniroma2.it (F.D.M.)

**Keywords:** Slipped capital femoral epiphysis (SCFE), In situ pinning (ISP), Hip remodeling after in situ pinning, Modified Dunn procedure (MDP), Hip instability (HI) following modified Dunn procedure (MDP), Pathogenesis of hip instability (HI)

## Abstract

Background. The modified Dunn procedure (MDP) has become popular during the last 16 years to treat severely displaced slipped capital femoral epiphysis (SCFE) while “in situ” pinning (ISP) has remained valid to treat mild to moderate SCFE, although the indication limit of the Southwick angle (SA) has not yet been established for either procedure. In this context, we reviewed two cohorts of patients with SCFE, one treated by ISP and the other by MDP. We also tried to better elucidate the etiopathogenesis of hip instability, a severe complication of MDP. Methods. Fifty-one consecutive patients with 62 hips affected by SCFE were treated by us from 2015 to 2019: 48 hips with a SA ≤ 40° had ISP while 14, with the SA > 40°, had MDP. The latter also had a CT scan to better investigate the SCFE morphology. Results were assessed using the Harris Hip Score. Results. The mean length of follow up of the two cohorts was 5.4 years (range: 3 to 8 years). Of the 35 hips operated by ISP with a full follow-up evaluation, 30 had an excellent or good result, 3, fair, and 2, poor. Of the 14 hips that underwent MDP, 11 had an excellent or good result, 1, fair, and 2, poor. A CT scan showed femoro-acetabular incongruency in two unstable hips following MDP. Conclusions. We performed ISP in chronic SCFE with the SA ≤ 40° and MDP in acute and chronic SCFE with the SA > 40°, with satisfactory results. In both acute-on-chronic and chronic long-lasting SCFE with severe displacement, planned for MDP, a CT scan should be carried out to evaluate possible femoro-acetabular incongruency that may cause hip instability.

## 1. Introduction

Slipped capital femoral epiphysis (SCFE) is caused by either acute or most frequently by progressive (chronic) displacement of the proximal femoral metaphysis [1] due to a failure of the proximal femoral growth plate that appears to be weakened by dysplastic changes [2,3]. Acute SCFE may also occur in chronic cases [1].

In chronic cases with mild displacement, i.e., with a Southwick angle (SA) ≤ 30°, “in situ” pinning (ISP) of the capital femoral epiphysis (CFE) halts disease progression with long-term satisfactory results [4] while, in cases with displacement from moderate (SA ranging from 31° to 50°) to severe (SA > 50°), early hip osteoarthritis frequently develops with poor results [5,6,7].

In acute cases treated according to the traditional methods of reduction and fixation, avascular necrosis (AVN) of CFE often occurs, leading to early hip osteoarthritis and disability [8,9].

Loder et al. [8] introduced the new concept of CFE stability to define acute and chronic SCFE. Acute cases are also defined as unstable, and patients are not able to stand and walk even with crutches, while in chronic cases CFE is stable, and patients are able to stand and walk albeit painfully and limping. 

During the last 16 years, the Dunn procedure [10] modified by Ganz (MDP) [11,12] has been gaining consent throughout the world to treat acute and chronic moderate to severe SCFE. In the hands of the authors who first described this new procedure [12]—compared to pinning “in situ” in chronic cases or to pinning after reduction with traditional methods in acute cases—MDP showed two main advantages: anatomic or nearly anatomic reduction of CFE and no AVN, with a potential reduction of late hip osteoarthritis [12]. 

Meanwhile, systematic reviews and meta-analyses of the literature on MDP have reported good results in most of the operated cases [13,14], and the first long-term follow-up studies have reported only a few cases of mild osteoarthritis after MDP [15,16,17]. However, as MDP has gradually been performed by a larger number of surgeons throughout the world, AVN, the most feared complication of MDP that was absent in the first reports by Leunig et al. [12], has increased to a mean ranging from 10.8% [13] to 14.3% [14]. Hip instability (HI), the second most feared complication, has been reported at a mean of 6.8% [13], although its etiopathogenesis has not been fully clarified.

In this study, we report the results of treatment in two cohorts of patients with SCFE, one including cases with the SA up to 40° operated on by ISP, and the other including cases with an SA > 40° operated on by MDP. We also attempted to better clarify the pathogenesis of HI that occurred in two of our cases operated on by MDP.

## 2. Materials and Methods

From January 2015 to December 2019, we treated 51 consecutive patients with 62 hips affected by SCFE.

Forty-eight hips affected by chronic SCFE with the Southwick angle (SA) ≤ 40° in 38 patients (10 patients had bilateral SCFE) were pinned “in situ”. Thirty-four hips had an SA ≤ 30° while in 14 hips, the SA measured a mean of 36° (range: 32° to 40°). Neither acute nor acute-on-chronic SCFE with SA < 40° were admitted to our hospital during the 5 years of the study. Informed consent was given by the parents of all the patients included in the study. The operation was performed on a traction bed, and one cannulated screw was used for “in situ” pinning (ISP). After surgery, the patients lay in bed for 1 day, and they were encouraged to move the operated hip as soon as they recovered from general anesthesia. Walking with crutches without weight-bearing on the affected hip was allowed for 8 weeks, while full weight-bearing was allowed at the 12th week after surgery. Of the 38 patients who had ISP, 28 with 35 operated hips came to the hospital for both clinical and radiographic evaluation, with a mean length of follow up of 5.3 years (range: 3 to 8 years with standard deviation of ±1.9) (Table 1). Six patients with 8 operated hips (2 patients with bilateral involvement) were not able to come because they lived far away from the hospital, but they had a telephone interview. Four patients with 5 operated hips (1 patient with bilateral involvement) were lost to follow up because they had changed their residency and could not be traced.

Fourteen hips affected by either acute or chronic or acute-on-chronic SCFE, with an SA > 40° (22.5% of all the treated hips) in 13 patients, were operated on by the modified Dunn procedure (MDP) (Table 2). The operation was performed by two senior surgeons (E.I. and E.M.) who were familiar with surgical dislocation of the hip as described by Ganz [11]. The diagnosis of chronic, acute, and acute-on-chronic forms was made based on the clinical data and plain radiographs. Patients with chronic stable SCFE were able to walk while patients with acute unstable SCFE were not. However, in some cases both the anamnestic and the radiographic data made it difficult to differentiate acute from acute-on-chronic SCFE.A CT scan with 3D reconstruction of the hips was also carried out in every patient to better define SCFE pathology. Measurements of both the acetabulum and the capital femoral epiphysis (CFE) were taken by using CT multiplanar reconstruction technique [18], and then visualized in the 3D reconstructed images to clearly show their anatomic location. Informed consent was obtained from the parents of all the patients included in the study. The operation was carried out according to Leunig et al. [12] (Figure 1A,B). After reduction, the capital femoral epiphysis (CFE) was fixed with 3 Steinmann pins, and the greater trochanter by 2 cannulated screws. After surgery, the patients lay in bed for 1 week, and were encouraged to move the operated hip. Thereafter, they were allowed to walk with crutches without weight-bearing on the operated hip for 5 weeks. Then, progressive weight-bearing with crutches was allowed to gradually reach full free weight-bearing 12 weeks after surgery if no sign of CFE osteonecrosis was evident on the hip radiographs carried out at that time. All the operated patients came to the hospital for both clinical and radiographic evaluation, with a mean length of follow up of 5.5 years (range: 3 to 8 years with standard deviation of ±1.6).

At follow up, the clinical history of the patients was collected starting from the operation to the time of follow up, and all the patients had both clinical and radiographic evaluation with an AP view of the pelvis and hips and a Lauenstein frog-leg view of the hips. A CT-3D scan of the hips was also carried out in patients who had had MDP and who showed radiographic signs of postoperative HI. The alpha angle was measured on the frog-leg view of the affected hips: 60° was considered the cut off between a normal hip and a hip with cam morphology [19]. The results of patients treated by either ISP or MDP were evaluated with the adult’s Harris Hip Score [20] rather than with other children’s hip scales, since all our hips had their growth plates closed at follow up. Complications in both cohorts were also reported. Descriptive statistics consisted of the mean ± SD for parameters with normal distributions after confirmation with histograms and the Kolmogorov-Smirnov test.

## 3. Results

All the data, results, and complications of the 28 patients operated on by “in situ” pinning (ISP) who came to the hospital for follow-up evaluation are reported in Table 1. Of the 25 hips with an SA ≤ 30° before ISP, 23 had an alpha angle < 60° at follow up while in two hips the alpha angle was >60°. Of the 10 hips with an SA > 30° and ≤40° before ISP, seven had an alpha angle < 60° while in three, the alpha angle was >60°. All the hips which showed good remodeling following ISP had open triradiate cartilage. Of the 35 operated hips, 19 had an excellent result (Figure 2A–D), 11 good, 3 fair and 2 poor. Two hips developed post-operative chondrolysis and both had a poor result. Three out of the 5 hips with an alpha angle > 60° had a fair result, showing clinical signs of femoro-acetabular impingement (FAI), and an arthroscopic osteoplasty was planned for those cases. Of the six patients (eight hips) interviewed by telephone, five had no pain and no functional limitations while one had slight pain after a long walk or sport activities.

All the data, results, and complications of the patients operated on by the modified Dunn procedure (MDP) who came to the hospital for follow-up evaluation are reported in Table 2. Of the 14 operated hips, eight had an excellent result (Figure 3A–D), 3 good, 1 fair and 2 poor. We had one case of massive AVN in a hip with acute-o- chronic SCFE, and two cases of hip instability, one in acute-on-chronic and one in chronic SCFE. The case with a fair result had a mild subluxation, while in the two cases with a poor result, one had a massive AVN (Figure 4A–D) and the other a marked subluxation of the hip (Figure 5A–E): both hips were salvaged by total hip replacement. In three hips, small asymptomatic heterotopic ossifications were present around the greater trochanter (Figure 3D).

Of the two patients with hip instability (HI), one had bilateral SCFE: the right hip had chronic SCFE with an SA of 42° while the left one had acute-on-chronic SCFE with an SA of 85°. Pain had been present for 8 months in the left hip and for 3 months in the right one, but SCFE was diagnosed only after admission to the hospital for the acute slipping on the left. The pre-operative CT-3D scan of the pelvis and hips carried out at admission to the hospital showed that the right capital femoral epiphysis (CFE), although postero-medially displaced, was fully contained into the acetabulum while the left CFE had a postero-medial intracapsular location (Figure 6A). The right acetabulum was almost hemispherical while the left one was elliptical, according to their orthogonal diameters (Figure 6B). The right CFE was almost hemispherical like the right acetabulum, while the left CFE had an irregular hood shape and it was thicker than the right one, appearing not congruent with the left acetabulum (Figure 6C). The other patient had a unilateral left chronic SCFE with a history of pain lasting for almost 1 year (Figure 7A). The CT-3D scan showed that the left acetabulum had an elliptical shape in comparison to the normal right one that was almost hemispherical, according to their orthogonal diameters (Figure 7B). The transverse CT scan, cut just above the acetabular notch, showed a perfect fit of the right CFE into the acetabulum while the left CFE lay far from the inner wall of the acetabulum after its surgical reduction by MDP (Figure 7C). The coronal CT scan showed that the left acetabulum was shallow and the left subluxated capital femoral epiphysis elliptical-shaped (Figure 7D).

## 4. Discussion

In 1964, Dunn [10] proposed a surgical technique for the open reduction and fixation of the capital femoral epiphysis (CFE) at the site of the slipping, and 14 years later Dunn and Angel reported the long-term results of the operation [21]. They recommended posterior osteophyte resection and mild shortening of the femoral neck before “gentle” reduction and fixation of CFE with three K-wires. The operation was carried out without dislocation of the hip. Particular attention was paid to not injure the “posterior synovial membrane” containing the retinacular vessels. In the authors’ experience, the prevalence of AVN was 24% in acute cases and 4% in chronic cases. However, other authors have not been able to reproduce Dunn’s good results in the following years, reporting a high complications rate [22,23,24].

Before the introduction of MDP, acute unstable cases of SCFE were treated by either closed or open reduction and pinning according to the traditional techniques [1]. AVN was a frequent major complication, the risk of which was affected by early intervention [8] and joint decompression [25]. To decrease AVN incidence, the ideal time for treatment is within 24 h from the beginning of symptoms, but if this is not possible, treatment should be delayed for one week. This interval of time during which treatment is not recommended is defined as the “unsafe window” [8]. Joint decompression is performed by perioperative arthrotomy that reduces the vascular tamponade due to the increased intracapsular hip pressure caused by hemarthrosis [25].

In 2001, Ganz et al. [11] described the “surgical dislocation” of the hip with all the tips and tricks to preserve the integrity of the retinacular vessels, while in 2007 Leunig et al. [12] applied the technique defined “modified Dunn procedure” (MDP) to the open reduction and fixation of moderate and severe SCFE, reporting anatomic reduction of CFE and 0% prevalence of AVN. Recent systematic reviews and meta-analyses of the literature on MDP [13,14] have shown overall satisfactory results, but with a mean prevalence of AVN ranging from 10.8% to 14.3%. However, the prevalence of AVN was almost double in acute cases (19.9%) [14], as confirmed by another recent meta-analysis on acute SCFE treated by MDP [26].

The gradual spread of MDP throughout the world during the last 16 years has changed the approach to SCFE treatment, prompting more and more surgeons to adopt MDP, attracted by the good result obtained with this technique. However, most authors have indicated MDP in moderate to severe SCFE [16,27,28,29,30,31,32,33,34,35,36,37,38,39,40,41,42,43,44], while others have extended the indication to mild SCFE as well [15,45,46,47]. This new trend has also influenced the indication for “in situ” pinning (ISP) that represents a milestone in SCFE treatment. Consequently, most authors have indicated ISP only in mild cases with the Southwick angle (SA) < 30°, while a few have extended the indication limit beyond 30°, also reporting satisfactory results [5,6,48,49,50]. Therefore, the upper SA limit for ISP as well as the lowest SA limit for MDP are still debated. 

The indication for SCFE treatment should be based on the few long-term follow-up studies on SCFE [6,48,50,51,52]. Those studies have shown that mild, as well as many moderate cases of SCFE, either untreated [51,52] or following ISP [6,48,50], will have in adulthood either no sign or mild to moderate signs of hip osteoarthritis, while cases with severe slipping will develop early, as will severe hip osteoarthritis. The latter is triggered by the conflict between the anterior part of the femoral neck that slips anteriorly and the antero-superior border of the acetabulum. That condition has been called femoro-acetabular impingement (FAI) [53], and it is assessed by the alpha angle, the normal value of which is still debated although a recent meta-analysis fixes it as ≤60° [19]. Undoubtedly, ISP blocks further slipping regardless of its severity, but it has no effect on the remodeling of the bump present at the capital femoral epiphysis (CFE)-neck junction: remodeling of the CFE-neck junction is thus crucial to avoiding FAI and the consequent hip osteoarthritis.

In a classic study, Jones et al. [54] showed that remodeling following ISP is strictly dependent on both the amount of the original metaphyseal displacement measured by SA and the patient’s residual skeletal growth potential represented by the activity of the triradiate cartilage. In that study, remodeling of the head-neck junction occurred in 90% of the cases with SA ≤ 30°and in 75% of the cases with SA ≤ 40°, provided that the triradiate cartilage was still open. In our study, of the 25 hips with chronic SCFE and SA ≤ 30° operated on by ISP and evaluated radiographically at follow up, 23 remodeled (92%), ending up with an alpha angle < 60°, and only one of the two cases with an alpha angle > 60° showed clinical signs of FAI. Lack of remodeling was observed in three out of the 10 patients with SA > 30° and ≤40°. The three patients had closed triradiate cartilage and an alpha angle > 60°. Those two parameters are at present negative predictors of the cervico-cephalic bump remodeling. Some other measurable extra criteria are hoped to be introduced in the future to predict cases with SA up to 40° that will not remodel. Moreover, not all cases with a cam morphology of the hip (i.e., alpha angle > 60°) will develop clinical signs of FAI, as already reported [19]. We had clinical evidence of FAI in only 3 out of 6 cases with an alpha angle > 60°; in those few cases, arthroscopic osteoplasty of the head-neck junction was advised [55,56]. Overall, the result of ISP was either excellent or good in 86% of the hips of our patients that were evaluated at follow up with the Harris Hip Score. In addition, five of the six patients who were interviewed by telephone reported more than satisfactory results.

Before the introduction of MDP, the poor outcomes of moderate to severe SCFE were salvaged by intertrochanteric and subtrochanteric femoral osteotomies, described by Imhauser in Europe [57] and by Southwick in North America [4]. Since the osteotomy site is distally far from the deformity, there are limits to the amount of deformity correction. Moreover, the osteotomy creates a secondary Z-shaped deformity at the femoral neck-shaft junction that can make future total hip replacement operations difficult. Both osteotomies are also technically demanding, and early severe complications like AVN and chondrolysis may occur, although long-term follow-up studies reported good results in almost 70% of cases [58,59].

In our small series of SCFE operated on by MDP, we had almost 90% of excellent-good results, in line with the current literature. AVN occurred in 7% of cases while hip instability (HI), in 14%; these are the reverse of the numbers reported by the recent systematic reviews and meta-analyses of the literature [13,14]. However, our statistics are biased by the low number of our patients who underwent MDP.

Treatment options are then based on two groups of SCFE differentiated by the cut off of SA alone. This seems to be a simplification of the problem since it is likely to suppose that other still unknown factors might influence the choice of treatment.

The etiopathogenesis of AVN in SCFE was clarified many years ago [60] while that of HI, described as a “devastating complication” by Upasani et al. [61], has not yet been fully elucidated. Various etiopathogenetic mechanisms have been formulated; some of them have been objectively confirmed, while others have been only theorized. Upasani et al. [61] reported HI in 4% of cases in 406 MDP in a multicenter study across eight institutions. They theorized a double cause of the postoperative HI: (1) the loose closure of the anterior capsule to prevent vascular insult; (2) previous damage to the anterior labrum done by the prominent anterior edge of the femoral neck. However, we always closed the anterior capsule loosely in our 14 hips, but HI developed in only two of our cases! 

Aprato et al. [62] reported three cases of HI following MDP. Two cases were caused by flattening of the acetabular roof, as shown by radiographs and CT scan, while the third case was caused by excessive shortening of the femoral neck (1 cm) to decompress the retinacular vessels. In this case, a concurrent cause was the re-fixation of the greater trochanter without distal advancement. Aprato’s findings are supported by a radiographic study of Maranho et al. [63], who found that 25% of patients with SCFE tended to develop decreased acetabular coverage featuring hip dysplasia. Favoring factors were young age at the initial slip, severe slipping, and CFE overgrowth. The authors’ conclusion was that acetabular remodeling during the last SCFE stages can be a favoring factor of HI after MDP.

After Aprato’s study, several authors have reported eight additional cases of HI [17,40,41,44] with a prevalence of about 6.6%. However, the etiology of HI was recognized in only two of the eight reported cases: one case was caused by a loose body into the hip joint [41] and the other by the interposition of the in-folded capsule between the femoral head and the acetabular walls [43].

Therefore, the previous literature shows that a trivial cause of HI was found in a few cases, and they had a good outcome after re-operation; in others, acetabular dysplasia due to acetabular remodeling was recognized as the cause, and they were fixed by periacetabular osteotomy; in still others, reduction of the subluxation was possible, and immobilization in a hip spica cast stabilized the hip: capsulo-labral insufficiency was suspected, but not objectively demonstrated. However, in yet other cases, no objective cause could be detected. Subluxation or dislocation could not be reduced, and in many of those cases secondary AVN developed with very poor outcomes [61].

In our two cases of HI, CT-scan studies showed why the subluxation could not be reduced: the deformed CFE could no longer fit completely into the acetabulum because their respective shapes had been modified during the SCFE’s evolution. In fact, both cases had long-lasting chronic and severe slipping that caused progressive remodeling of both the acetabulum and the CFE in accordance with their new relationships caused by the severe slip: the CFE was only partly contained into the acetabulum since half of it was located within the postero-medial capsular fold. Consequently, the open triradiate cartilage remodeled the acetabulum in relation to the new position of the CFE that kept growing misshapen outside of full acetabular containment since its vascular supply was still feeding it.

We measured both CFE and the acetabulum with the method defined as “CT multiplanar technique” described by Wang et al. [18]. This method allows the operator with a dedicated computer program to simultaneously take the measurements on the three spatial planes of a skeletal component that had CT carried out with 3D reconstruction.

We then might speculate that femoro-acetabular incongruency, like that of our two cases, could also have been the cause of HI in previously reported cases in which an objective cause was not found by clinical and radiographic means. 

Our study has several limitations. It is a retrospective cohort study subject to associated bias with the study design. The MDP sample size is small. A prospective study involving a larger number of patients is needed in the future to assess the validity of this novel surgical procedure. However, the minimal and mean follow up of our MDP patients were of sufficient length to provide medium-term data, and no patient was lost to follow up. Additionally, we focused exclusively on SA to establish the cut off between ISP and MDP. Therefore, the results might be affected by the method of measurement although SA has been classically used to classify the severity of SCFE since its first description [4]. Other methods of classification of the slip severity have been proposed including the calcar femorale (CF) angle, but the reliability for both intra- and inter-observer measurements of both SA and CF method are very similar [64]. Finally, the number of ISP patients with SA > 30° and ≤40° was less than half of those with SA ≤ 30°. A larger number of the former is needed in the future to support our indication to ISP for patients with SA up to 40°.

In conclusion, ISP is our preferred method of treatment for mild to moderate chronic SCFE with SA ≤ 40°. The few cases that might be complicated later by symptomatic FAI can be treated by arthroscopic osteoplasty with satisfactory results and almost no complications [55,56]. In chronic cases with SA ≥ 40° in which early osteoarthritis of the hip is very likely to develop, we prefer MDP, but the patient’s family must be informed of the risk of a severe complication like AVN ranging from 10% to 14% [13,14]. We also prefer MDP in acute SCFE owing to the reduction of the risk of AVN. In fact, the incidence of AVN is about 19.9% with MDP [26] while it is about 23.9% [9] with peaks up to 47% [8] with the traditional techniques of reduction and pinning. Unfortunately, the salvage treatment in most of the cases of AVN is total hip replacement burdened with its well-known negative drawbacks when performed at a very young age [65]. Concerning the etiopathogenesis of HI, one of the feared complications of MDP, we recommend performing a CT scan study in long-lasting SCFE cases with marked displacement to evaluate femoro-acetabular incongruency as a possible cause of post-operative HI.

## Figures and Tables

**Figure 1 children-10-01163-f001:**
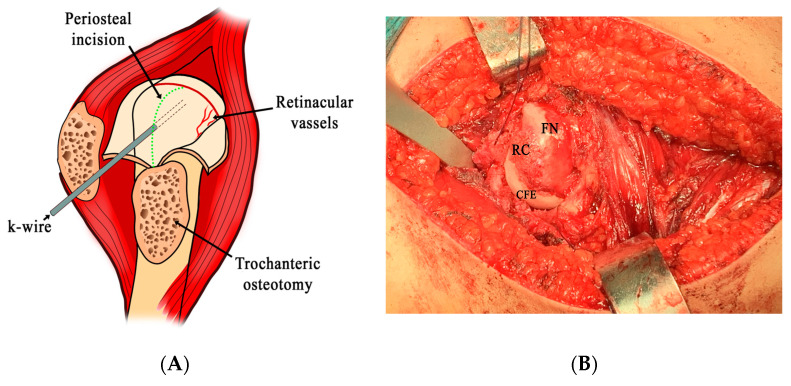
(**A**) Scheme of the surgical approach of the modified Dunn procedure after trochanteric flip osteotomy, Z-shaped capsulotomy, and surgical dislocation of the hip. The green dotted line indicates the posterior osteotomy of the greater trochanter, and the red line indicates incision of the periosteum of the femoral neck to obtain the periosteal flap containing the retinacular vessels. The K-wire fixes the femoral capital epiphysis before dislocation in unstable slips. (**B**) Surgical dislocation in a chronic stable SCFE. FN: femoral neck; RC: reactive callus; CFE: capital femoral epiphysis.

**Figure 2 children-10-01163-f002:**
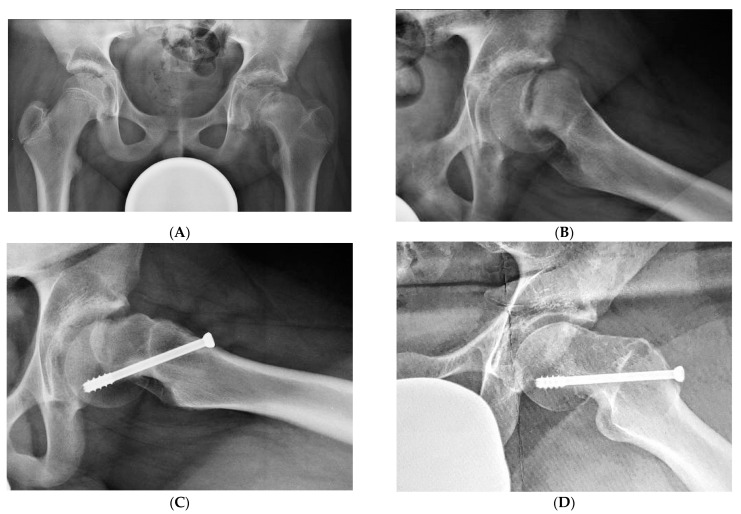
(**A**) Antero-posterior and (**B**) Lauenstein view of a chronic SCFE with a Southwick angle of 40° in a 13.2-year-old boy with open triradiate cartilage. (**C**) Eight months after fixation “in situ” with 1 cannulated screw. (**D**) At follow up, 6 years later, the patient was asymptomatic. The Lauenstein view showed a remodeled neck-head junction and an alpha angle of 62°. The result was rated as excellent, with a Harris Hip Score of 96.85 points.

**Figure 3 children-10-01163-f003:**
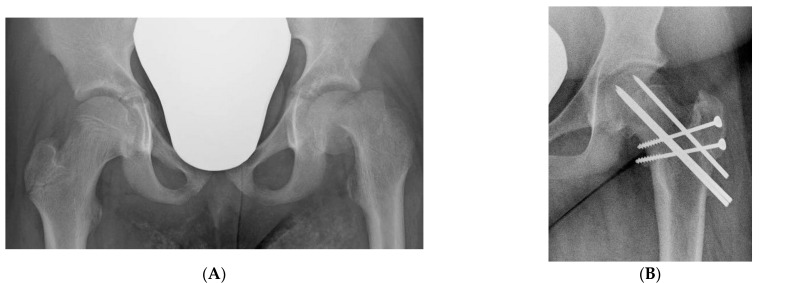
(**A**) Antero-posterior and (**B**) Lauenstein view of an acute-on-chronic SCFE with a Southwick angle of 70° in an 11-year-old girl. (**C**) Open reduction by modified Dunn procedure and stabilization of the capital femoral epiphysis with 3 K-wires. (**C**) At follow up, 6 years later, the patient was asymptomatic. The antero-posterior view of the pelvis showed the left femoral neck shorter than the right one and small heterotopic ossifications at the greater trochanter. (**D**) The alpha angle measured 54° on the Lauenstein view, and the result was rated as excellent with a Harris Hip Score of 97 points.

**Figure 4 children-10-01163-f004:**
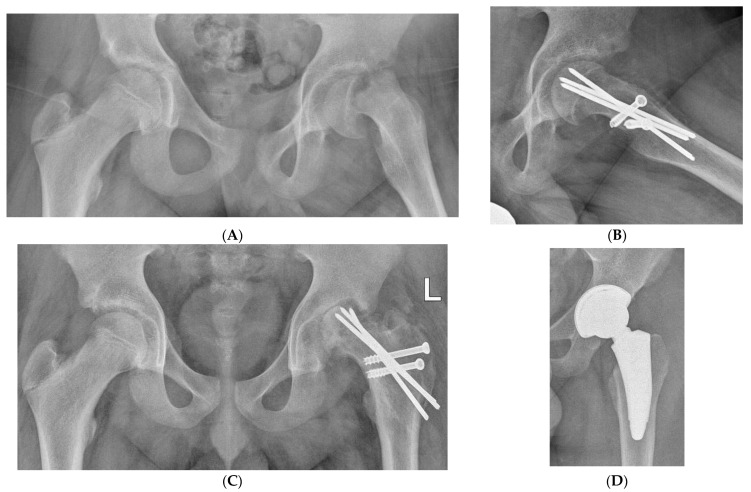
(**A**) Antero-posterior view of left acute SCFE with a Southwick angle of 58° in a 13-year-old boy. (**B**) Lauenstein view showing anatomic reduction by modified Dunn procedure and stabilization of the capital femoral epiphysis with 3 K-wires. (**C**) Five months later, avascular necrosis of the capital femoral epiphysis developed. (**D**) A total hip replacement was performed 3 months later.

**Figure 5 children-10-01163-f005:**
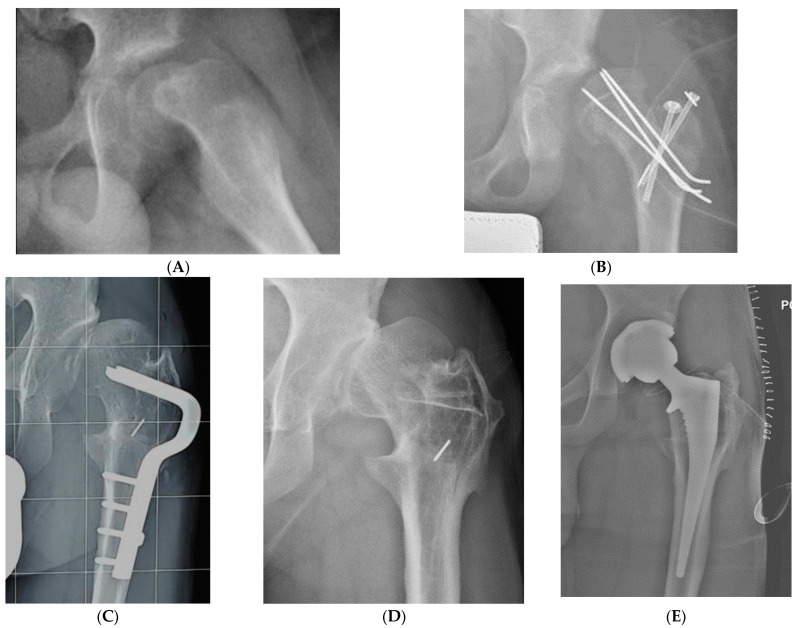
(**A**) Lauenstein view of an acute-on-chronic SCFE with a Southwick angle of 85° in a 14-year-old boy. (**B**) Marked instability was evident immediately after surgery. All the attempts at reduction failed, including a proximal femoral osteotomy. (**C**) Four years after the femoral osteotomy, hip subluxation persisted but the patient was able to walk without pain. (**D**) Eight years after the modified Dunn procedure, a severe symptomatic osteoarthritis developed, and (**E**) a total hip replacement was performed.

**Figure 6 children-10-01163-f006:**
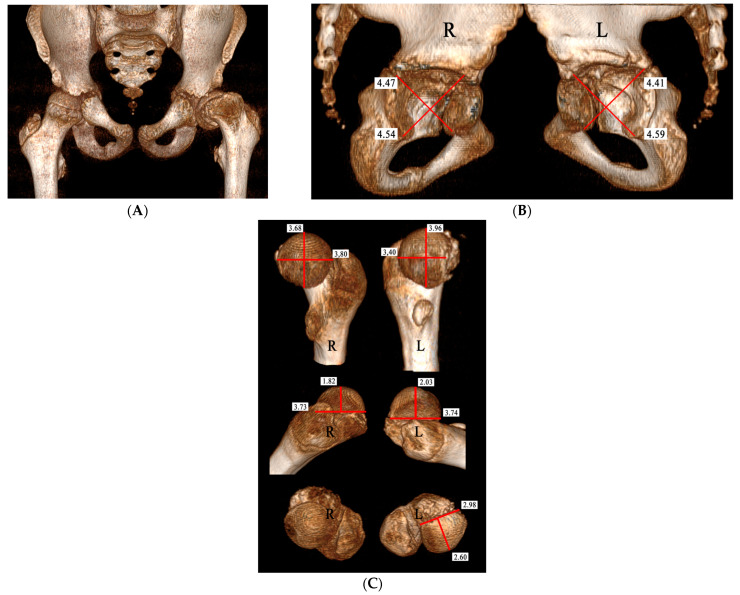
(**A**) Pre-operative CT scan with 3D reconstruction of the pelvis and hips of the patient with acute-on-chronic SCFE on the left and chronic SCFE on the right, whose left hip is illustrated in Figure 5. The left capital femoral epiphysis was located within the postero-medial capsular fold. (**B**) The right acetabulum (R) is irregularly hemispherical while the left one (L) is rather elliptical, according to the length of their orthogonal diameters. (**C**) The 2 capital femoral epiphyses are oriented parallel to the coronal plane at the top part of the figure, parallel to the transverse plane (viewed from posteriorly) at the middle part while, at the bottom, they are viewed from above with the anatomical axis of the proximal femur perpendicular to the coronal plane. The right epiphysis is almost hemispherical, matching the shape of the right acetabulum, while the left one has an irregular hood-like shape and is thicker than the right one (the measurements are in centimeters). A mismatch between the capital epiphysis and the acetabulum appears to be the cause of the left hip instability illustrated in Figure 5.

**Figure 7 children-10-01163-f007:**
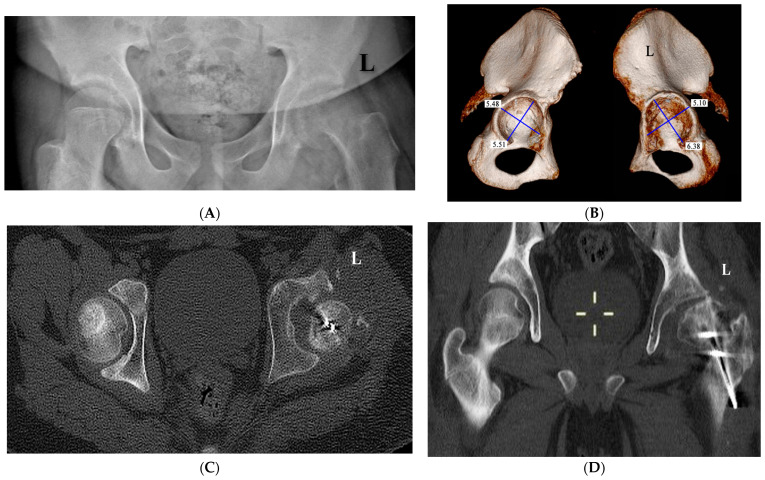
(**A**) Antero-posterior view of a severe left chronic SCFE in a 14.2-year-old boy who had been symptomatic for 1 year. (**B**) CT scan with 3D reconstruction of the pelvis and hips shows that the left acetabulum (on the right side of the figure) has an elliptical shape while the right one is irregularly hemispherical (the measurements are in centimeters). (**C**,**D**) CT scan of both hips after modified Dunn procedure performed on the left. On the transverse plane (**C**), the right capital femoral epiphysis fits perfectly into the acetabulum while on the left, the capital epiphysis is smaller than the right one and lies far from the inner wall of the acetabulum. On the coronal plane (**D**), the left acetabulum is shallow and the left subluxated capital femoral epiphysis is elliptical-shaped.

**Table 1 children-10-01163-t001:** Demographics, clinical and radiographic data, results, and complications of the 28 patients with 35 hips affected by chronic SCFE with the Southwick angle ≤ 40°, operated on by “in situ” pinning, who had clinical and radiographic follow-up evaluation.

Patient Number	Age (Years) (Mean ± SD)	Gender	Hip Number and Side	Southwick Angle (Mean ± SD)	Length of Follow Up (Years) (Mean ± SD)	Age at Follow Up (Years) (Mean ± SD)	Alfa Angle (Mean ± SD)	Results (Harris Hip Score)	Complications
28	11.7 (±1.7) (range: 9.6 to 14.2)	Male (16) Female (12)	Right (10) Left (11) Bilateral (7) Total (35)	27.3° (±7.1°) (range: 16° to 40°)	5.3 (±1.9) (range: 3 to 8)	16.1 (±1.8) (range: 12.6 to 22.2)	52.7° (±7.9°) (range: 40° to 66°)	Exellent (19 hips) Good (11 hips) Fair (3 hips) Poor (2 hips)	Chondrolysis (2 hips) Femoro-acetabular Impingement (3 hips)

**Table 2 children-10-01163-t002:** Demographics, clinical and radiographic data, results, and complications of the 13 patients with 14 hips affected by either acute or acute-on-chronic or chronic SCFE with the Southwick angle > 40°, operated on by the modified Dunn procedure, who had clinical and radiographic follow-up evaluation.

Patient Number	Age at Surgery (Years) (Mean ± SD)	Gender	Hip Number and Side	Classification	SOUTHWICK Angle (Mean ± SD)	Length of Follow Up (Years) (Mean ± SD)	Age at Follow Up (Years) (Mean ± SD)	Alfa Angle (Mean ± SD)	Results (Harris Hip Score)	Complications
13	12.9 (±0.8) (range: 11.5 to 14.2)	Male (11) Female (2)	Right (4) Left (8) Bilateral (1) Total (14)	Chronic (7 hips) Acute/chronic (3 hips) Acute (4 hips)	72.7° (±12.2°) (range: 42° to 85°)	5.5 (±1.6) (range: 3 to 8)	18.2 (±2.4) (range: 14.5 to 22.2)	44.7° (±5.6°) (range: 38° to 52°)	Exellent (6 hips) Good (5 hips) Fair (1 hip) Poor (2 hips)	AvascularNecrosis (1 hip) Instability (2 hips) Heterotopic Ossifications (3 hips)

## Data Availability

The data supporting the results of this study are property of the Bambino Gesù Pediatric Hospital. For reasons of privacy and adherence to ethical guidelines, these data are not freely accessible. Specific requests to access the data may be addressed to the Bambino Gesù Pediatric Hospital, in compliance with internal policies and authorization procedures.

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
