# Peer review of "Changing Treatment Philosophy of Slipped Capital Femoral Epiphysis (SCFE) after Introduction of the Modified Dunn Procedure (MDP): Our Experience with MDP and Its Complications"

_children, 2023, doi:10.3390/children10071163_

Round 1
Reviewer 1 Report
The paper reflects current shift of the paradigm of treatment of SCFE, in the direction of splitting the indication for the specific treatment according to the amount of displacement in two basic variants: in situ pinning, and sub-capital osteotomy, mostly in the variant of modified Dunn procedure. Basically the authors confirm their commitment to the previously published approach that mild slipping should be an indication to pinning and moderate and severe slipping - to MDP. The question remaining is in regards to the cut-off point for the decision-making. As the author stated, general recommendation is to use Southwick angle 30° as the cut-off point for choosing one or another method. The authors present their consecutive series of 51 patient with 62 hips, affected by SCFE treated with pinning or MDP. There are two major research questions, postulated by the authors in this particular study. The first question is in concern of extension of the cut-off point for decision-making between two procedures. The authors used Southwick angle less than 40° as the indication. Using the statistical analysis of the whole group of patients with Southwick angle less than 40° operated by pinning they conclude that the results are comparable with the previously reported and in favour of the usage of this technique for patients with less than 40° of Southwick angle. With this statement the authors recommend this cut-off point for the decision making. It should be noticed that the methodology for this conclusion does not look very strong. The authors do not provide detailed comparative analysis of the two groups - with SA, less than 30°, and with the same angle of 30 to 40°. According to the data presented in the text of the paper, this comparison does not look so undoubtedly supportive for the extension of the indication to 40°. If the general statistics of good results within this group is 92% (good remodelling), in 10 cases with Southwick angle from 30 to 40° only 70% hips did remodel. The authors emphasise that other indicators of the following remodelling like alpha angle and the open triradiate cartilage are the prerequisites for the remodelling as well, so if the authors assumed 30 to 40° of slipping as acceptable for the pinning, it is logical to introduce some measurable extra-criteria in presence of almost 1/3 of poor remodelling in this group. If so, this conclusion should be better structured in the text of the paper as well as in the formal conclusion. And ideally should be better statistically supported. The second major findings of the authors regards to the hip instability - disabling complication of SCFE. The authors speculated the potential mechanism of aetiology and pathogenesis of this condition. They postulate that severely misshaped and secondary deformed by slipping femoral head and opened triradiate cartilage are the fundamental prerequisites for the secondary instability. This interesting finding is illustrated by the CT scans of two cases with these changes. As the conclusion and practical recommendation, the authors strongly recommend to use CT scans for long-lasting SCFE cases with marked capital displacement to evaluate incongruency as the possible cause of post-operative instability. The authors used linear measurements of the transverse size of the femoral head and acetabulum to demonstrate misshaped components. Despite the interesting concept if the authors strongly recommend this assessment it is important to propose more specific recommendations for the measures. Otherwise this strong recommendation doesn’t lead to the practical effects. There are also some minor recommendations. It’s not recommended to use the expressions like “we believe” in the scientific paper. And some radiographs are presented upside down and should be reverted. The paper needs some revision according to the reviewer’s point of view.Author Response
"Please see the attachment."

Reviewer 2 Report
* The subject matter at hand possesses a reasonable level of clarity and has received considerably more attention and analysis than acknowledged in the present introduction. The article introduces a promising concept that holds merit. However, there are certain aspects of the study that give rise to a few concerns from my perspective.
* The study included a total of fifty-one consecutive patients, with a total of sixty-two hips affected by slipped capital femoral epiphysis (SCFE). The treatment was administered by our team over a period of fifteen years, from 2015 to 2019. Out of the sixty-two hips, forty-eight had a Southwick angle (SA) of less than 40° and were treated using in situ pinning (ISP). On the other hand, fourteen hips with an SA greater than 40° underwent modified Dunn procedure (MDP). Among these fourteen hips, sixteen also underwent a computed tomography (CT) scan to gain further insight into the morphology of SCFE.
* In addition to the aforementioned points, it is crucial to highlight the significance of including high-quality figures in the article. It would greatly enhance the overall presentation if the authors could provide high-resolution figures that accurately depict the data and findings. Regrettably, some of the figures currently included in the article suffer from a low resolution, which diminishes their clarity and impact.
* The conclusion drawn in the article is well-founded and supported by the methods employed and the results obtained.
* Add a section (Discussion). The authors don't discuss the limitations of this study. Please add it.
* Add a section (conclusion)
No comment
Author Response
"Please see the attachment."

Reviewer 3 Report
Thank you for letting me review this manuscript. Overall, it is relevant and well written. My comments are listed below.
Abstract
Line 12
“We attempted to ascertain SA indication values….”
the presented cohorts, albeit significant considering the diagnosis and treatment, indication values for SA cannot be ascertained. This is, however, not stated in the relevant and reasonable conclusion. The aim and the conclusion should have a clearer connection. I suggest re-phrasing the ambition.
Introduction
Line 32.
The epiphysis has not slipped (despite the SCFE acronym)- the main slipping is in the metaphysis. The epiphysis remains stationary in the acetabulum. There are multiple references for this.
This incorrect description is stated many times in the manuscript (e.g discussion 264-267)
Line 39.
The authors have decided to use the acute and chronic terminology. The differentiation between “stable” and “unstable” is, in my opinion, more relevant and increasingly used (Loder et al.). This terminology should at least be addressed.
Materials and Methods
Line 66-67
The regimen of weight-bearing regimen is a choice of the authors and there is no consensus regarding this. Some choose to allow weight-bearing based on the vascularity of the caput after surgery. Do the authors have any thoughts on this?
Line 84.
The differentiation between acute, chronic, and acute on chronic is, according to the authors, based on “the basis of the clinical data and plain radiographs”. In my experience this is not always clear cut. An acute may well be an acute on chronic. A little more information on how this was done by the authors is needed as well as a recognition that this may be challenging.
Line 109
Which patients were “selected” for this ?
Line 112
The Harris Hip Score was developed for adult patients. Why is not e.g Children's Hospital Oakland Hip Evaluation Scale (CHOHES) used? This should be discussed.
Discussion
Line 251-262
This reasoning implies that there are only two groups of SCFE, based on any give cut-off of the SA angle, and that treatment options should only be based on this. This is, in my opinion, a simplification. The authors have grouped the patients in this study based on the SA angle. The MDP group consists of both chronic and acute while the ISP group only consists of chronic. Do the authors, for example, find any indication for reduction and pinning for unstable SCFE (and if so for which degrees of slippage?). This
There are also other factors and methods that may affect the risk for AVN, like decompressing the joint per-operatively. The authors do not mention these methods or factors
Limitations
There is no clear limitations section in the discussion which is warranted.
There are other methods to measure the severity of slip than the SA, e.g the calcar femorale (CF) angle. The results may be affected by the method used to measure and this should be mentioned in the discussion.
Author Response
"Please see the attachment."

Round 2
Reviewer 1 Report
The authors followed the reviewer’s comments and modified the paper according the recommendations. The paper is recommended for publication in the present version.